# Identification of Maize with Different Moldy Levels Based on Catalase Activity and Data Fusion of Hyperspectral Images

**DOI:** 10.3390/foods11121727

**Published:** 2022-06-13

**Authors:** Wenchao Wang, Wenqian Huang, Huishan Yu, Xi Tian

**Affiliations:** 1College of Physical Science and Information Engineering, Liaocheng University, Liaocheng 252000, China; wangwc0501@163.com; 2Beijing Research Center of Intelligent Equipment for Agriculture, Beijing 100097, China; huangwq@nercita.org.cn; 3College of Engineering, China Agricultural University, Beijing 100083, China

**Keywords:** maize, moldy level, catalase activity, hyperspectral image, data fusion, feature selection

## Abstract

Maize is susceptible to mold infection during growth and storage due to its large embryo and high moisture content. Therefore, it is essential to distinguish the moldy sample from healthy groups to prevent the spread of mold and avoid huger economic losses. Catalase is a metabolite in the growth of microorganisms; hence, all maize samples were accurately divided into four moldy grades (health, mild, moderate, and severe levels) by determining their catalase activity. The visible and shortwave near-infrared (Vis-SWNIR) and longwave near-infrared (LWNIR) hyperspectral images were investigated to jointly identify the moldy levels of maize. Spectra and texture information of each maize sample were extracted and used to build the classification models of maize with different moldy levels in pixel-level fusion and feature-level fusion. The result showed that the feature-level fusion of spectral and texture within Vis-SWNIR and LWNIR regions achieved the best results, overall prediction accuracy reached 95.00% for each moldy level, all healthy maize was correctly classified, and none of the moldy samples were misclassified as healthy level. This study illustrated that two hyperspectral image systems, with complementary spectral ranges, combined with feature selection and data fusion strategies, could be used synergistically to improve the classification accuracy of maize with different moldy levels.

## 1. Introduction

Maize is an important food crop, feed crop, and cash crop [1]. Compared with wheat and rice, maize has larger embryos, and its moisture content at harvest reaches about 30%, higher than the 25% and 22.5% of wheat and rice, respectively, which makes it more susceptible to mold infection during growth and storage [2]. Among them, *Aspergillus flavus* is the strain that most easily and commonly infects maize; aflatoxin (AFB1), produced by *Aspergillus flavus*, is extremely carcinogenic and toxic, and is the most toxic mold secondary metabolite in contaminated food [3,4]. Humans and animals eating food contaminated with AFB1 is a serious threat to life and health safety. Therefore, the early real-time detection of moldy maize has very important research significance.

Traditional detection methods for moldy maize include sensory evaluation and physical and chemical component detection [5]. Sensory evaluation is simple, time-saving, and low cost, but the evaluation results are easily disrupted by the external environment and the subjective emotions of the inspectors. Additionally, the toxic substances will also pose a threat to the health of the inspectors. The physical and chemical component determination is generally detected by high-performance liquid chromatography (HPLC), polymerase chain reaction (PCR), and enzyme-linked immunosorbent assay (ELISA) [6,7,8]. Although these methods can achieve more accurate measurement and qualitative analysis, they require expensive testing equipment and professional technicians, and the testing process is complex and time-consuming [9]. Hence, using traditional methods, it is difficult to achieve simple, rapid, and non-destructive detection, which cannot meet the actual needs for identification of kernel maize.

In recent years, non-destructive detection technologies, such as electronic nose, machine vision, near-infrared, and hyperspectral imaging, have been successfully applied to the classification of moldy maize [10,11]. Electronic nose technology [12] is mainly used specific sensors to identify the level of mold based on the change of volatile organic compounds (VOCs) information. Leggieri et al. [13] used electronic nose technology to determine the concentration of AFB1 and fumonisins (FBs) in maize; the prediction model of AFB1 and FBs built by the artificial neural networks were 78% and 77%, respectively. However, the concentration of gas is easy to change in the flow state, which affected the discrimination accuracy. Machine vision detection mainly adopts machine learning algorithms to extract features from kernel images and then establish the classification model based on extracted features [14]. Shi Ying [15] extracted R-channel eigenvalues of RGB images (red, green and blue three-channel color image) and classified maize kernel samples with different levels of mold using the Back Propagation (BP) neural network. Visible-near infrared (Vis-NIR) spectroscopy technology connects the spectral information with the internal content of substances and uses the spectral curve to analyze the changes of internal components of seeds in the process of mold growth [16]. Therefore, the machine vision and Vis-NIR spectroscopy can express the external and internal changes of the target samples respectively. However, both internal quality and external characteristics of maize will change during the moldy process; neither machine vision nor Vis-NIR spectroscopy can obtain internal and external quality information at the same time. Hyperspectral imaging technology combines spectral analysis technology with image processing technology, which can simultaneously obtain the spectral data with internal component information and the image data with appearance feature information, realizing the rapid, pollution-free and non-destructive detection [17,18,19].

In terms of using hyperspectral imaging technology to identify moldy maize, Tao et al. [20] used random frog (RF) combined with partial least-squares discriminant analysis (PLS-DA) to qualitatively analyze the healthy maize and polluted maize inoculated with aflatoxigenic fungus at different culture days based on long wave near-infrared (LWNIR) hyperspectral images; the classification accuracy of the calibration set and verification set was 82.3% and 94.9%, respectively. Williams et al. [21] evaluated the fungal development in maize kernels using LWNIR; principal component analysis (PCA) was firstly used to remove the interference of noise, such as background, bad pixels, and shadows, from the hyperspectral images. Three distinct clusters related to the degree of infection were found in the scoring plots of PC4 and PC5. Dai et al. [22] established a classification model of moldy maize with different culture days (0 days, 2 days, 4 days, 6 days, and 8 days) based on 9 characteristic wavelengths selected from visible and short wave near-infrared (Vis-SWNIR) hyperspectral imaging using fisher discriminant analysis (FDA); the classification accuracy of the calibration set and validation set were 100% and 98.67%, respectively, illustrating that the characteristic wavelengths could represent the main information about moldy levels of maize samples. Del Fiore et al. [23] used Vis-SWNIR hyperspectral imaging combined with multivariate statistical analysis to identify maize kernels infected with fungi under different growths. The results showed that hyperspectral imaging was able to quickly distinguish between healthy and infected maize, i.e., 48 h after inoculation with mycorrhizal fungi. Previous studies have shown that both the spectral ranges of Vis-SWNIR and LWNIR can be used to distinguish the moldy level of maize; however, it has not been found that fusing the spectral information of different hyperspectral systems can construct a classification model of maize with different moldy levels. Yu et al. [24] studied the influence of Vis-SWNIR and LWNIR hyperspectral imaging systems on the prediction ability of total volatile basic nitrogen (TVB-N) content in tilapia fillets during refrigeration; the results showed that the fused spectral data of both sensors achieved a better prediction result than that of individual sensor. Meanwhile, the study of fusing spectral data with texture data to discriminate the moldy level of maize is less extensive. Ma et al. [25] developed the classification model of fresh and frozen meats based on the spectral and texture information extracted from Vis-SWNIR hyperspectral images; the research showed that the classification model built by the feature fusion of spectra and texture was better than that of spectra and texture alone. Therefore, fusing the information of spectra and texture obtained from different hyperspectral image systems would be a new idea for constructing an accurate classification model of maize with different moldy levels.

Mold growth is uncontrollable. Hence, moldy levels may not be uniform among different samples at the same culture time and a small number of samples were not consistent with the designed moldy levels. Catalase (CAT) is a metabolite in the growth of *Aspergillus flavus* and other microorganisms [26]. Zhang et al. [27] found that the correlation coefficient between the number of mold colonies and the activity value of CAT reached more than 0.9 in various grains such as wheat, rice, and maize. In addition, CAT is the precursor product of AFB1 produced by *Aspergillus flavus*. Zhang et al. [28] analyzed the correlation between CAT activity value and AFB1 content of moldy maize and found that both of them had the same change curves. The above research showed that CAT activity value could reflect the moldy levels of maize. However, at present, it is not found that the model for discrimination of maize with different moldy levels was established based on the feature fusion of hyperspectral imaging information and CAT activity value of moldy maize.

In this study, we proposed a new method to better divide the maize with different moldy levels by monitoring the CAT activity value of maize samples infected with *Aspergillus flavus* under different culture days. The objective of this study is to examine the potential of using multi-levels data fusion of hyperspectral images to identify the maize with different moldy levels. The specific objectives of this study were to: (1) analyze the difference of spectra and texture of Vis-SWNIR and LWNIR hyperspectral images of maize samples with different moldy levels; (2) examine the ability of different pretreatment methods and classifier for identification of maize samples with different moldy levels; (3) compare the classification ability of the models based on the pixel-level fusion of spectra and different texture parameters; (4) evaluate the effects of features selected by different variable selection methods on the classification models of feature-level fusion; and (5) establish the best classification model of maize with different moldy levels by integrating the spectrum and texture data with Vis-SWNIR and LWNIR regions.

## 2. Materials and Methods

### 2.1. Maize Sample Preparation

“Zhengdan 958” is widely planted in China due to its advantages of high and stable yield; hence, “Zhengdan 958” was selected as the experimental maize sample of this study. To reduce the influence of the bacteria carried by the maize itself, maize kernels with the same size and appearance were selected manually. All maize kernels were surface sterilized by soaking in 2% sodium hypochlorite solution for 5 min and then rinsed three times with distilled water. The conidia suspension of *Aspergillus flavus* (BNCC142801 purchased from BeNa Biotechnology Research Center, Xinyang, Henan province, China) was diluted to 10^−3^ in sterile water and inoculated into maize kernels. Simulation of the maize mold process by inoculation with molds has been widely used in laboratory studies. Tao et al. [20] used two Aspergillus flavus, AF13 (aflatoxin-producing Aspergillus flavus) and AF36 (non-aflatoxin-producing Aspergillus flavus), for artificial laboratory inoculation to study the changes in maize. In this study, all inoculated maize kernels were divided into 240 groups and placed in petri dishes with the embryo side facing upward. Each group contained about 30 kernels, weighing 10 ± 0.5 g. All samples were cultured in a constant temperature and humidity incubator with a temperature of 30 °C and relative humidity of 80%.

### 2.2. Hyperspectral Image Acquisition System

The Vis-SWNIR and LWNIR hyperspectral reflectance imaging system (Figure 1) built in the Intelligent Detection Laboratory of China Agricultural Intelligent Equipment Technology Center, was used to acquire the hyperspectral images of moldy maize samples in the wavelength range 327–1098 (nm) and 930–2548 (nm). The Vis-SWNIR hyperspectral imaging system consists of an imaging spectrometer (ImSpector V10EQE, Spectra Imaging Ltd, Oulu, Finland), an electron multi-plying charge-coupled device (EMCCD) camera (Andor Luca EMCCD DL-604 M, Andor Technology plc., Belfast, UK) with a resolution of 502 × 500 and a camera lens (OLE23-f/2.4, Spectral Imaging Ltd., Oulu, Finland), and a spectraCube data acquisition software (Isuzu Optics Corp., Xinzhu, Taiwan, China) controls the operation of mobile platform and acquisition of hyperspectral images. The LWNIR hyperspectral imaging system consists of an imaging spectrometer (ImSpector N25E, Spectral Imaging Ltd., Oulu, Finland), a charge-coupled device (CCD) camera (Xeva-2.5-320, Xenics Ltd., Leuven, Belgium) with a resolution of 320 × 256 and a camera lens (HSIA-OLE22, Spectral Imaging Ltd., Oulu, Finland), spectral acquisition software (Isuzu Optics Corp., Xinzhu, Taiwan, China). The two hyperspectral acquisition systems shared two 300 w halogen lampsadjusted at an angle of about 45 to provide a stable light source, a motorized displacement stage (EZHR17EN, AllMotion, Inc., Union City, CA, USA) for sample placement, and a computer (Dell, Intel (R) Core (TM) i5-2400 CPU @ 3.10 GHz) with two types of hyperspectral acquisition software.

The moldy levels of maize varied with the culture time. In this study, to artificially cultivate the maize samples with different moldy grades, sixty samples were taken out from a constant temperature and humidity incubator on day 0, day 2, day 4, and day 6 as the moldy samples at healthy, mild, moderate, and severe levels, respectively. In order to collect the high-quality images without saturation and distortion, hyperspectral images of samples were acquired by line scanning method, and the distance between the lens to the moving platform, the exposure time and the mobile platform speed were set to 430 mm, 3 ms, 2.6 mm/s for Vis-SWNIR hyperspectral systems and 310 mm, 5 ms, 40 mm/s for LWNIR hyperspectral systems, respectively.

### 2.3. Determination of CAT Activity

CAT activity value could reflect the activity strength of mold. Therefore, CAT activity value of the maize samples at four moldy levels was determined using the potassium permanganate titration method [29] after hyperspectral image collection. The specific steps of CAT activity determination were as follows:

Step 1: Weigh the sample and place it in a conical flask;

Step 2: Add 40 mL distilled water and 5 mL 0.3% hydrogen peroxide, and set another control group (40 mL distilled water and 5 mL 0.3% hydrogen peroxide in an empty conical bottle), and shake them in a shaker for 20 min;

Step 3: Add 5 mL 3 mol/L sulfuric acid, shake for 5 min, take it out, filter with filter paper, take out 10 mL filtrate, titrate with 0.005 mol/L potassium permanganate solution until light pink, and do not change color for 30 s.

The value of CAT activity (Equation (1)) is expressed as the volume (*mL*) of 0.005 mol/L potassium permanganate consumed in unit weight (*g*) and time (*h*). Where $V_{1}$ and $V_{2}$ were the titrated potassium permanganate volume (*mL*) of the control group and sample group, M was the mass (*g*) of the sample, and T was the unit time (*h*).
(1)$$Value(\frac{mL}{g\times h})=\frac{\left(V_{1}-V_{2}\right)}{M\times T}$$

### 2.4. Hyperspectral Image Processing and Information Extraction

The original hyperspectral image needs to be corrected to eliminate the influence of light source and camera dark current changes [30]. The standard white reference image was acquired using a white Teflon plate (99% reflectivity) under the same sampling environment as the sample. Turn off the light source and cover the lens to obtain a black reference image (0% reflectivity). The corrected image is calculated using the black and white reference image by Equation (2):(2)$$I_{C}=\frac{I_{O}-I_{B}}{I_{W}-I_{B}}$$
where $I_{O}$ was the original hyperspectral image, $I_{W}$ and $I_{B}$ represented the white reference and black reference images, respectively, and $I_{C}$ was the corrected hyperspectral image. In this study, the hyperspectral image correction and subsequent data processing were performed in MATLAB 2019B (The MathWorks, Inc., Natick, MA, USA).

To extract the information of region of interest (ROI), the mask method was used to segment the target and background of the corrected hyperspectral images. The gray images at 849 and 1098 nm were used to construct a binary mask by setting appropriate thresholds, because the spectral intensity difference between the gray image background and maize was largest at 849 and 1098 nm wavelength images for Vis-SWNIR and LWNIR hyperspectral images, respectively. Then, the corresponding hyperspectral image was multiplied by the filtered mask to remove the background information. The original and denoised RGB images of maize with different mold levels were shown in Figure 2. After acquiring the ROI region, the average spectrum was extracted from all pixels of ROI region for each wavelength of the hyperspectral image. Due to the noise and useless information in the beginning and end bands, a total of 389 spectral variables within 399–1001 nm and 112 spectral variables within 1005–1701 nm were obtained from the hyperspectral images of Vis-SWNIR and LWNIR regions, respectively.

The extraction of texture features was realized by the gray-level co-occurrence matrix (GLCM). The GLCM described the probability of occurrence of two pixels with different distances and directions in a gray image [31]. The data of four texture parameters (contrast, correlation, energy, and homogeneity) in each ROI band was extracted from the GLCM, by Equation (3)–(6). In this study, the pixel distance was set as 1, and only the GLCM in the four directions of 0°, 45°, 90°, and 135° was considered. The average value in the four directions was used to describe each texture parameter characteristic. After removing the noise bands, four texture parameters feature matrices with sizes of 240 × 389 (240: number of samples; 389: number of variables) and 240 × 112 were obtained in the Vis-SWNIR and LWNIR bands, respectively.
(3)$$contrast=(\sum_{i=1}^{N}\sum_{i=1}^{N}{\left(i-j\right)}^{2}P(i,j))$$
(4)$$correction=\frac{\sum_{i=1}^{N}\sum_{i=1}^{N}\left(ij\right)P(i,j)-\mu_{i}\mu_{j}}{\sigma_{i}\sigma_{j}}$$
(5)$$energy=\sum_{i=1}^{N}\sum_{i=1}^{N}P{\left(i,j\right)}^{2}$$
(6)$$homogeneity=\sum_{i=1}^{N}\sum_{i=1}^{N}\frac{P(i,j)}{1+{\left(i-j\right)}^{2}}$$
where (i,j) was the pixel coordinate, P(i,j) was the joint probability with two neighboring pixels, and N (set N = 8 in this research) was the number of gray-levels. $\mu_{i}$, $\mu_{j}$, $\sigma_{i}$, and $\sigma_{j}$ represented the mean and standard deviation of the row and columns in the GLCM, respectively.

### 2.5. Spectral Data Preprocessing

The hyperspectral data were easily interfered by random noise, stray light, background, and equipment in the hyperspectral images acquisition. To eliminate the influence of environmental factors and improve the correlation between spectral data and chemical composition, it was necessary to preprocess the raw spectrum [32]. Hence, moving smooth, multiple scattering correction (msc), detrend, and mean centralization (center) were used in this study. Studies have indicated that the smooth was used to remove noise interference in the spectrum and improve the signal-to-noise ratio. Its basic idea is to smooth the raw data through the “averaging” or “fitting” of several points in a finite size spectral window. The spectral window size must be an odd number, and the wider the window, the lower the spectral resolution. Msc used the method of least squares to fit the linear relationship between each spectrum and the average spectrum. This means that msc could eliminate scattering bias. Detrend is an approach to eliminate the baseline drift in the spectrum and the influence of different sampling batches on the spectrum. Firstly, a trend line was derived from spectral values and wavelengths through least squares fitting, and then the trend line was subtracted from the original spectrum. The center was effective in enhancing the differences between data, its basic idea is to remove the column, row, or overall average from each column, row, or both separately [33,34,35]. In this study, smooth was firstly used to reduce the noise and interference existed in original spectra, and then msc, detrend, and center were employed secondly to process the spectra on the basis of smooth. The best spectral preprocessing method was determined by comparing the effects of different pretreatment methods on classification accuracy, then the spectral data processed by the best method were fused with texture information for further analysis.

### 2.6. Data Fusion

Data fusion was a process of combining information from different independent information sources, which could express the described objects or processes in more detail and complete than using a single information source alone. Generally, data fusion was divided into pixel-level fusion, feature-level fusion, and decision-level fusion according to the fusion level from low to high [36]. In the study, the spectral data and texture data obtained from hyperspectral images within Vis-SWNIR and LWNIR regions were fused at pixel-level and feature-level respectively for developing a high accuracy and robust classification model of moldy maize.

Pixel-level fusion was simply merging the data information of different sources [37], so the fused features contained more variables, which was conducive to further data processing. However, it could also input the irrelevant and redundant variables into the model. In this study, pixel-level fusion models were built by fusing the spectra matrices with texture parameters, for Vis-SWNIR and LWNIR regions. For the new matrix formed after data fusion, one row represented the characteristic information of the same sample, and one column represented the eigenvalues at a specific wavelength. The optimal combination of spectrum and texture parameters was obtained by evaluating the classification accuracy of the developed models.

Feature-level fusion was to extract features from a single data block using the variable selection method and then integrate the processed feature matrix [38,39]. Compared with pixel-level fusion, feature-level fusion could adjust the number of features from different data, especially when there were large differences between single data blocks. In this study, both spectral matrix and texture matrix obtained from Vis-SWNIR and LWNIR ranges had 389 and 112 variables, respectively, these data were often multicollinearity and redundant, especially between adjacent bands. Therefore, feature wavelength selection was commonly adopted to select the key wavelengths from full-band data, which could reduce redundant and noisy information, as well as simplify the model. The optimal combination of spectral data and texture parameters in pixel-level fusion was used as the data source of feature-level fusion. Three kinds of variable selection methods including variable combination population analysis (VCPA) [40], iteratively retains informative variables (IRIV) [41], and hybrid method mVCPA-IRIV [42] were used to select the features that carry the information of moldy maize from the spectral and texture parameters data, and then combined them into a new data matrix to build the feature-level model. The number of columns in the new data matrix was the number of features obtained from the two data.

### 2.7. Discriminant Model and Evaluation

In order to obtain an accurate and reliable classifier, the original spectral data were used to establish a classification model with SVM, Random Forest (RF), and K-nearest neighbors (KNN). The performance of different classifiers was compared, and the optimal classifier method was selected to be used in the subsequent data processing classification algorithm.

The basic idea of SVM was to divide the segmentation hyperplane with the maximum classification interval according to the training samples in the feature space. When facing the nonlinear problem, the kernel function was introduced and transformed into a linear problem in high-dimensional space through nonlinear transformation. SVM was often used for problems with a small sample set or linear indivisibility [43]. The radial basis function (RBF) kernel function had more advantages in dealing with the nonlinear relationship between feature information and categories [44], hence RBF was selected as the kernel function of SVM in this study. The optimal loss parameter and kernel parameter was searched by the cross-verification grid optimization method.

RF was an ensemble learning method based on the Bagging algorithm, which could be used to solve classification and regression problems. RF had the advantages of processing high-dimension data, strong adaptability to data sets, and fast training speed [45]. In this study, when the RF classifier was trained, the number of decision trees was set to 50 to store the observation results of each tree.

KNN was a commonly used classification algorithm. Its core idea was to select k nearest neighbor samples in the feature space. In these K samples, if most samples belong to a certain category, the test samples also belong to this category [46]. In this study, when the KNN algorithm was used for training, parameters were automatically optimized to obtain the optimal nearest neighbor number and distance measurement parameters.

The rationality of data set division affects the prediction performance of the classification model. To avoid the influence of artificially selected calibration prediction sets on the results, in this study, all 240 samples were sequentially divided into 4 moldy levels based on the determined CAT activity value, so there were 60 samples in healthy, mild, moderate and severe levels, respectively. Then, the 60 samples of each category were randomly divided into calibration and prediction sets with a proportion of 3:1. Hence, 180 samples were selected as the calibration set to build the calibration model, and the 60 remaining samples were selected as the prediction set for evaluating the performance of the established model.

The performance of the model was evaluated from four aspects: classification accuracy of the calibration set and prediction set, and overfitting. Generally, a good model should have higher classification accuracy and lower differences between calibration and prediction sets. The main key steps of this study were shown in Figure 3.

## 3. Results and Analysis

### 3.1. CAT Activity Analysis of Maize with Different Moldy Levels

Table 1 shows the range, mean, and standard deviation of CAT activity values under different moldy levels. The maize kernels used in this experiment were sterilized, and the CAT activity value of the sample was 0 under the healthy level. The results showed that CAT activity values increased with the aggravation of maize moldy levels, and CAT activity values increased rapidly in the early stages of mold and slowly in the late stages, which was related to the growth pattern of the mold. The CAT activity values of different moldy levels had obvious gradient differences, indicating that CAT activity values could be used to determine the level of moldy maize.

### 3.2. Spectral and Texture Characterization

The curves of original spectra and texture data extracted from the hyperspectral images of Vis-SWNIR and LWNIR regions were shown in Figure 4. The solid lines and the shaded part of Figure 4a,b are the average spectra and the standard deviation of maize with different moldy levels, respectively. Figure 4c–j shows the texture data of contrast, correlation, energy, and homogeneity extracted from the ROI of all samples. It was clear that both spectra data and texture data of maize with different moldy levels had similar trends, but their reflectance intensity were significantly different, which may be related to the decomposition of chemical substances in the process of maize mold.

By analyzing the spectral curve characteristics, it can be easily found that the more serious maize mold, the lower the spectral reflection intensity in both wavelength regions, indicating that the light absorption capacity of mold tissue was stronger than that of maize tissue. The spectral curve is monotonous in the Vis-SWNIR region, the average spectral curve gradually increased in the region of 399–820 nm and then decreases slowly. However, the spectral curve was complicated and varied in the LWNIR region. Two obvious reflectance peaks were captured around 1100 nm and 1300 nm, respectively. The former may be related to C-H in lipids [47], and the latter can be designated as a combination between the first overtone of N-H stretching with the fundamental N-H in-plane bending and C-N stretching with N-H in-plane bending vibrations [48]. In addition, there were two obvious absorption peaks at 1192 nm and 1445 nm. The peak at 1192 nm may be associated with the second overtone of C-H stretching in carbohydrates [49] and at 1445 nm may be related to the O-H bond in water and the first overtone of C-H in protein [50]. There were significant differences in the reflectance spectra and texture intensity between different moldy levels. These differences may provide the possibility of classifying the maize with different moldy levels. However, the spectra of maize samples with different moldy levels crossed in some wavelength intervals (1400–1701 nm), and there was no significant correlation between the reflectance spectra and the moldy levels. Hence, the spectra and textures should be fused to research the classification ability of their latent information.

### 3.3. Comparison and Optimization of Different Classifiers and Preprocessing Methods

In non-destructive detection technology based on visible and near-infrared hyperspectral images, many spectral preprocessing methods and classifiers can be used to construct the classification model. In order to improve the development efficiency of the classification models, the classifier and spectral preprocessing method were firstly determined in this study based on the original spectral data. Three classification algorithms, including SVM, RF, and KNN, were used to build the classification model based on the original spectra extracted from different hyperspectral systems. Table 2 shows the CAT activity value of the calibration and prediction datasets. It should be pointed out that the samples in the calibration and prediction sets were kept unchanged and no single sample was used in calibration set and prediction set at the same time. The results of different classification algorithms were shown in Figure 5. Compared the overall classification accuracy yielded by different classifiers, SVM was the most robust classifier with the accuracy of calibration and prediction sets of 93.89% and 86.67%, and 86.67% and 85% for Vis-SWNIR and LWNIR regions, respectively. Hence, SVM was used as the only classifier to build the classification models in the subsequent data processing. In terms of the classification accuracy built by the SVM models, the LWNIR region was poorer than the Vis-SWNIR region, which may be due to the weak correlation between the spectral intensity and the level of mold in the 1400–1701 wavelength range, as can be seen by the spectral characteristic curve.

In terms of spectral preprocessing methods, to reduce the noise and interference in the original spectral information, the 9-point smooth method was firstly used to eliminate the noise existed in the original spectra. Then msc, detrend, and center methods were carried out on the basis of smoothed spectra to further optimize the spectral data, and the influence of different preprocessing methods on classification accuracy was compared. The average spectra preprocessed by different method were shown in Figure 6a–f. It can be seen that these preprocessing methods effectively eliminate signal offset and light scattering. The classification results of different preprocessing methods were shown in Table 3. For the case of the Vis-SWNIR region, the best preprocessing method was smooth-detrend with the classification accuracy of 86.67% and 88.33% for the calibration set and prediction set, respectively, because the accuracy gap between the calibration set and prediction set was the smallest. For the case of the LWNIR region, the model developed by smooth-detrend spectra achieved better prediction performance with the classification accuracy of 90.56% and 88.33% for the calibration set and prediction set, respectively. All classification models developed by the spectra preprocessed by smooth-detrend were superior to the model developed by original spectra, proving that spectral processing could greatly improve the reliability of classification models. The detrend could eliminate the influence of different sampling batches on the spectrum and improve the robustness and accuracy of the classification model. Sanchez et al. [35] improved the prediction ability of strawberry quality parameters by using detrend spectral pretreatment method. Paz et al. [51] found that detrend pretreatment had better prediction effect on sugar content and hardness in plums. Furthermore, the method of spectral pretreatment depends largely on the analyte being modeled and must be based on the judgment of the analyst [52]. In this study, smooth-detrend was selected as the most optimal spectral preprocessing method, and the spectra pretreated by smooth-detrend preprocessing were used for subsequent analysis instead of the original spectra in both Vis-SWNIR and LWNIR regions.

### 3.4. Pixel-Level Fusion Based on Full Wavelengths Spectra and Texture Data

Spectra and texture data carried the component content and component distribution information of the target sample, respectively. In order to develop a higher accuracy classification model, four texture parameters, including contrast, correction, energy, and homogeneity, were extracted and used to create a new fusion matrix by fusing with the spectral data at the pixel-level level. Then the most optimal combination of spectra and texture was determined by establishing classification models based on the fused data.

The classification results of the pixel-level fusion of spectral and texture information were shown in Table 4. Compared with the results obtained by individual spectral data, the classification ability of fusion data varied with the participation of different texture features. The texture feature of energy and contrast had a positive effect on improving the classification models, with the accuracy of prediction sets of 90% and 90% for Vis-SWNIR and LWNIR regions, respectively. In general, the contrast parameter reflects the clarity of the image according to the depth of the texture groove, and the energy parameter reflects the randomness of the image texture. The amount of mold increased with the increase of cultured time, and the mold mainly concentrated in the embryo region of the maize, which may be the reason why both texture features were more conducive to the classification of maize with different moldy levels. It should be pointed out that, other combinations of spectra and texture features had not yielded the desired results, suggesting that the prediction ability of pixel-level fusion was not the accumulation of data quantity. Although the pixel-level fusion of spectra and textures directly merged the data of different sources, this method could input valuable information to the model, but it can also add a large number of uncorrelated and noisy variables, resulting in the fused data could not significantly improve the predictive power of the model. Similar results were obtained using NIR and ATR-FTIR data blocks to detection of adulteration in honey [39], models based on a data matrix generated by pixel-level data fusion show no significant improvement in accuracy.

### 3.5. Classification Model Built by Feature-Level Fusion of Spectra and Texture Data

The classification of maize with different moldy levels based on hyperspectral imaging technology involved the rapid collection of a substantial number of hyperspectral images, which were composed of two spatial dimensions and one spectral dimension data. Then, the spectral or texture information was extracted from these hyperspectral images and used to predict the categories of each maize sample. A large number of spectral or texture variables in the full wavelength range often contained noise from the environment and instrumental sources, leading to complexity and poor predicting ability of a calibration model. In addition, when used for online or at-line purposes, the complex calibration models developed with the whole spectrum will not be applicable. To resolve these issues, the features of spectra and textures were extracted alone using variable selection algorithms including VCPA, IRIV, and mVCPA-IRIV; then, the classification model of feature-level was established based on the extracted features for Vis-SWNIR and LWNIR regions, respectively.

In order to analyze the features selected from spectral and texture data, characteristic wavelengths selected by the same variable selection method were concatenated together and their distribution maps were shown in Figure 7. Comparing the number of selected characteristic bands, the VCPA selected fewer variables than IRIV and mVCPA-IRIV for spectral and texture data in both Vis-SWNIR and LWNIR regions. The number of selected characteristic bands was greater in texture data than that of spectral data for all three variable selection methods, illustrating that the texture had more information about the moldy maize than that of spectral data. Comparing the distribution of characteristic bands selected by three variable selection methods, it could be found that many common regions were determined in both spectral and texture data. The shared regions were concentrated at 629–649, 728–743, 764–772, 855–860, and 1055–1248 nm for the spectral data (Figure 6a,c), while the shared regions were observed to be concentrated at 410–490, 584–592, 679–693, 866–876, 953–963, 1029–1060, 1167–1192, 1235–1267, and 1688–1701 nm for the texture data (Figure 6b,d). According to previous studies, Stasiewicz et al. [53] classified different levels of Aspergillus flavus in maize at 850 nm near 857 nm. Moreover, 768 nm and 853 nm were used to differentiate the fungal contaminated maize from healthy samples in Chu’s study [54]. Furthermore, 1029–1267 nm belonged to the second overtone of N-H stretching of proteins, as well as C-H stretching in lipids [47,48]; 1688–1701 nm was attributed to the second overtone of S-H, these were associated with protein, fat and starch [55]. Mold growth broke down the fat, protein, and starch of the maize kernels, which would change the reflection spectra and textures features of maize kernels, and resulting in the selection of the above characteristic bands. Except for those shared regions, there were some differences in the selected characteristic bands, which may be caused by the different principles of variable selection methods.

The classification results of the feature-level fusion of spectral and texture information were shown in Table 5. It could be found that the feature-level fusion models achieved better accuracy and reliability in Vis-SWNIR range than that of LWNIR region. In particular, the feature-level fusion models of VCPA, IRIV, and mVCPA-IRIV increased by 3.33%, 5.00%, and 1.67%, respectively, compared to the model based on pixel-level fusion for Vis-LWNIR region (Table 4). In detail, the prediction accuracy of the model based on the features selected by VCPA, IRIV, and mVCPA-IRIV was 93.33%, 95%, and 91.67% for Vis-SWNIR, and 90%, 83.33%, and 91.97% for LWNIR region. Although the IRIV method achieved the best prediction results in Vis-SWNIR region, the prediction ability was very poor in LWNIR region. In addition, the IRIV was time-consuming in variable selection and the number of selected variables used for modeling was much larger than that of the VCPA approach. However, the VCPA method achieved the classification accuracy of 93.33% and 90% for Vis-SWNIR and LWNIR regions, which was more robust than IRIV and mVCPA-IRIV. It is worth noting that, as a hybrid variable selection method, mVCPA-IRIV did not yield the best prediction results either in the Vis-LWNIR or in the LWNIR region. This result was consistent with the results of employing two-step hybrid methods to determine TVB-N contents in tilapia fillet for single Vis-NIR and NIR data blocks [24]. This may be because feature selection was based on a single data block with a relatively small number of variables, which was not suitable for the hybrid variable selection methods.

In conclusion, feature-level fusion model on the basis of variable selection had a large potential for distinguishing the maize with different moldy levels than the model of pixel-level fusion with full-band in both Vis-SWNIR and LWNIR regions. Remarkably, in the study by Anting et al. [37], the feature-level-PCA strategy using sample image and the spectra to evaluate the fermentation degree of black tea had a similar result with our proposed strategy.

### 3.6. Determination of the Optimal Feature-Level Fusion Model

Currently, the above results were based on two separate hyperspectral systems; meanwhile, typical hyperspectral imaging systems rarely extended the wavelength range of 399–1701 nm. Generally, more detailed and comprehensive feature information can be acquired from a wider spectral range, which makes sense given that integrating spectral and corresponding texture parameter data from Vis-SWNIR and LWNIR systems. The feature-level fusion model built by characteristic variables selected by VCPA obtained the most reliable classification result, hence, these variables (9 spectral features and 12 energy features for the Vis-SWNIR region, and 12 spectral features and 12 energy features for the LWNIR region) were integrated to build the classification model. The overall prediction accuracy was 96.11% and 95% for calibration and prediction sets, respectively, which was greater than the pixel-level fusion model and feature-fusion model of independent sensor. Although the number of variables used for modeling increased, it was far lower than the number of variables for full wavelength data. Similar results were also obtained in the internal bruising detection of blueberry by combining two hyperspectral systems with feature fusion strategies [56]. Figure 8 shows that the overall predicted results for the maize with different moldy levels. Except for the moderate level, all the moldy maize groups reached a high accuracy of more than 95%. In particular, all healthy maize was correctly classified. Some moderate levels samples were misclassified as mild or severe levels, resulting in a classification ability of only 90% for moderate levels, which agreed with the result of Yao et al. [57]. The moldy maize at moderate levels was difficult to accurately identify, which may be caused by the reduced variation between different moldy levels. This phenomenon also could be found through the determination of the CAT activity; with the aggravation of moldy level, the increase of CAT activity value among different categories decreased. However, it was worth emphasizing that none of the moldy samples were misclassified as healthy level, illustrating that the classification model had a certain practicality and objectivity.

By comparing previous studies, some non-destructive testing techniques to identify grain mold have been studied extensively. These single technologies, such electronic nose [13], machine vision [15], Vis-SWNIR hyperspectral systems [22,23], and LWNIR hyperspectral systems [20,21] have been used to monitor the health condition of maize during storage. Remarkably, these studies obtained satisfactory results. Due to the growth and multiplication of mold, both the internal quality and external characteristics of maize change during the moldy process. Hence, the strategy of using single technologies to evaluate the quality of maize was limited. In our study, the spectral and different texture parameter data were extracted based on the collected Vis-SWNIR and LWNIR hyperspectral images. The data fusion strategy significantly improved the richness of information, which was helpful for building a robust classification model. Therefore, it can be concluded that feature-level fusion model based on spectral and texture information of two hyperspectral systems could be used to improve the classification accuracy of maize with different moldy levels.

## 4. Conclusions

As an important grain crop, maize is susceptible to mold infection during growth and storage due to its large embryo area and high moisture content. Therefore, it is essential to distinguish the moldy sample from healthy groups to prevent the spread of mold and avoid huger economic losses. Hyperspectral imaging technology combines spectral analysis technology with image processing technology, which can simultaneously obtain the spectral data with internal component information and the image data with appearance feature information, realizing the rapid, pollution-free, and non-destructive detection. In this study, the hyperspectral images of maize with different moldy levels were collected within Vis-SWNIR and LWNIR regions, and the spectra and texture information were extracted and used to establish the classification model with the methods of pixel-level and feature-level fusions. The results showed that data fusion strategies at both levels achieved better classification results than spectra alone. For pixel-level data fusion of spectral and texture information, the energy and contrast achieved positive effect on improving the classification model, with prediction accuracy of 90% and 90% for Vis-SWNIR and LWNIR regions, respectively. The improvement in model detection accuracy is not very apparent, as some irrelevant variables are introduced along with useful information. For feature-level data fusion of spectral and texture information, the variables selected by VCPA significantly increase the classification accuracy, with prediction accuracy of 93.33% and 90% for Vis-SWNIR and LWNIR regions, respectively. Feature-level fusion models based on the key variable combination of two hyperspectral systems were best for the classification of maize with different moldy levels, with an overall prediction accuracy of 95.00% for each moldy level.

This paper mainly focused on the identify of moldy maize; it should be noted that the data fusion strategies presented in this study are generally suitable to the quality detection of other grain crops such as wheat, rice, and peanut. Although this study had shown that the great feasibility of using hyperspectral imaging technology and multi-source data fusion method to discriminate the maize with different moldy levels, there must be some differences between naturally and artificially moldy maize samples, and we will use the data fusion strategies to classify moldy maize under natural growth in our future work. Additionally, AFB1 is a metabolite of mold with high toxicity; therefore, the growth monitoring of mold plays an important role in the early warning of AFB1 pollution in maize. CAT is a precursor product synthesized by AFB1, its dynamic activity reflects the level of mold activity, and it has a significant relationship with the content of AFB1. Therefore, we will develop a model for warning the AFB1 contamination based on the relationship between CAT of mold and AFB1 under different moldy conditions, which provides theoretical basis and technical guarantee for safe storage of maize.

## Figures and Tables

**Figure 1 foods-11-01727-f001:**
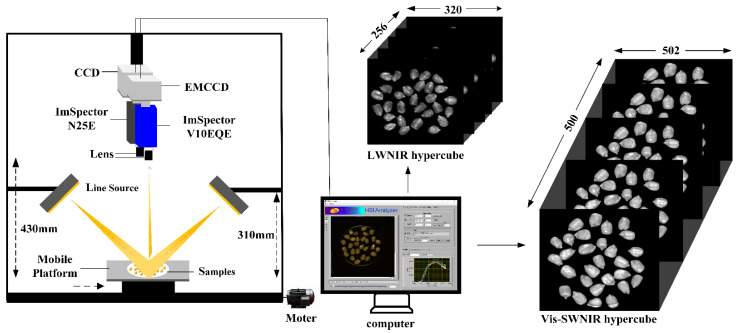
Hyperspectral image acquisition system.

**Figure 2 foods-11-01727-f002:**
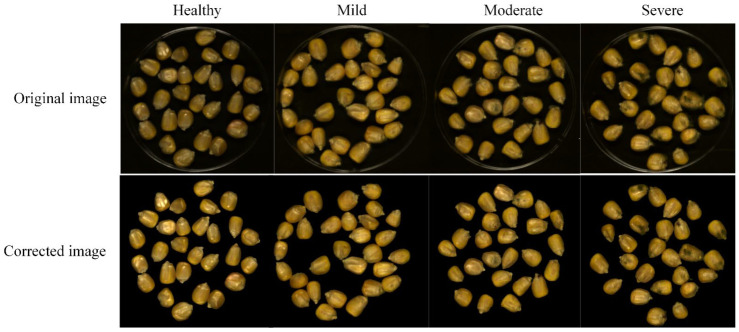
Original and denoised RGB images (red, green and blue three-channel color image) of maize with different mold levels.

**Figure 3 foods-11-01727-f003:**
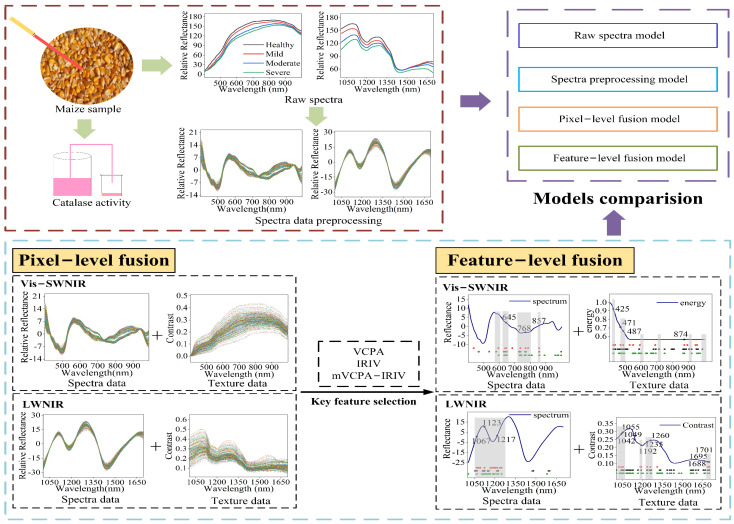
The experimental scheme of the data fusion model for identification of maize with different moldy levels.

**Figure 4 foods-11-01727-f004:**
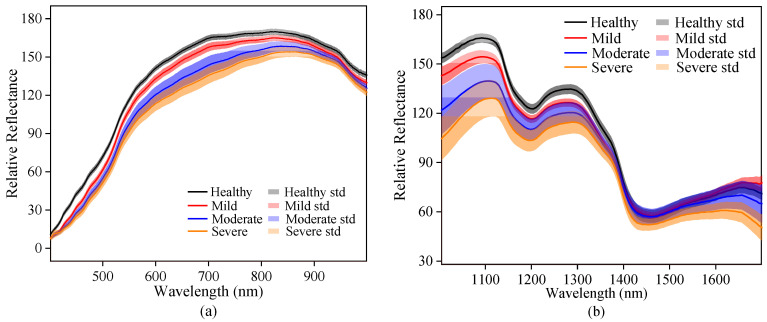
Spectra and texture curves of maize with different moldy levels in the Vis-SWNIR region (left) and LWNIR region (right): (**a**) original spectra in Vis-SWNIR region; (**b**) original spectra in LWNIR region; (**c**) contrast parameter in Vis-SWNIR region; (**d**) contrast parameter in LWNIR region; (**e**) correction parameter in Vis-SWNIR region; (**f**) correction parameter in LWNIR region; (**g**) energy parameter in Vis-SWNIR region; (**h**) energy parameter in LWNIR region; (**i**) homogeneity parameter in Vis-SWNIR region; (**j**) homogeneity parameter in LWNIR region.

**Figure 5 foods-11-01727-f005:**
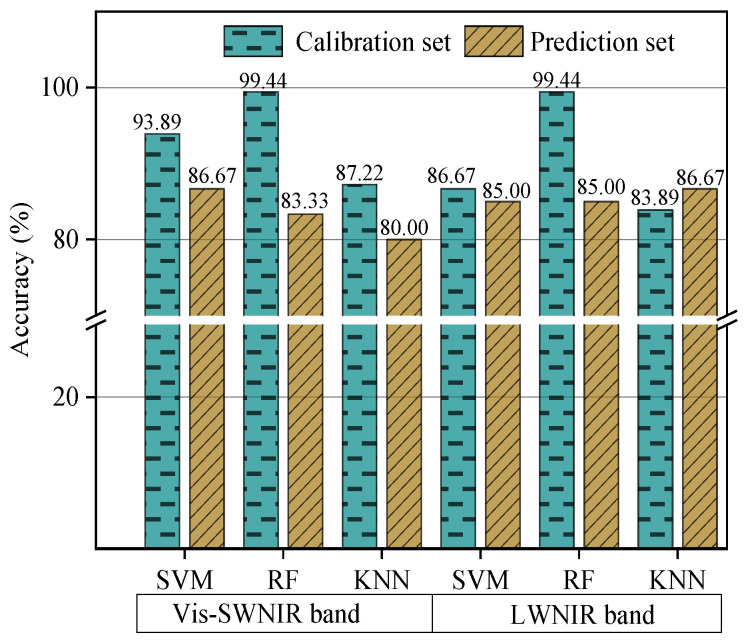
The classification results of different classifiers based on the original spectra of Vis-SWNIR and LWNIR hyperspectral systems.

**Figure 6 foods-11-01727-f006:**
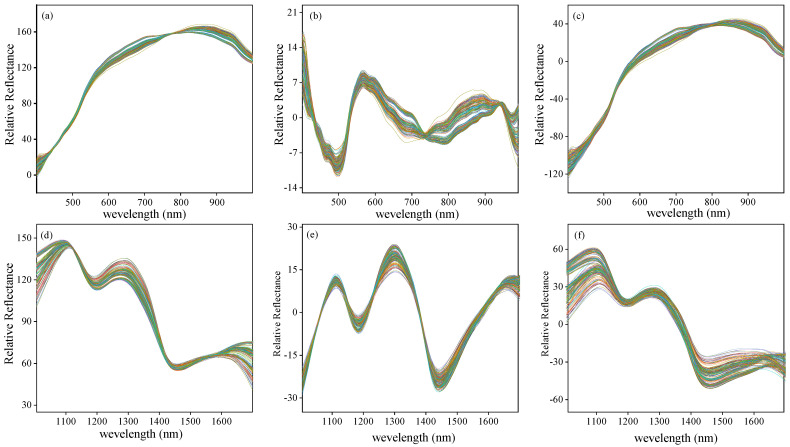
The preprocessed spectra by smooth-msc (**a**,**d**), smooth-detrend (**b**,**e**), and smooth-center (**c**,**f**) methods for the Vis-SWNIR and LWNIR regions, respectively.

**Figure 7 foods-11-01727-f007:**
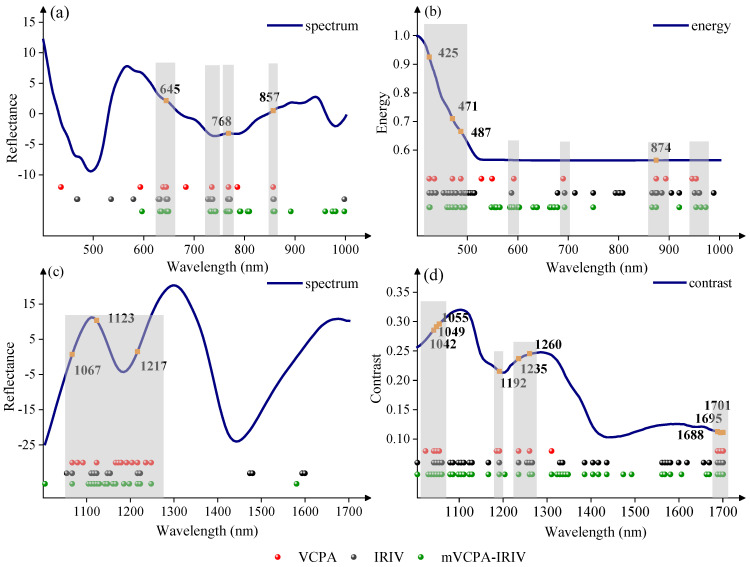
The distribution of key wavelengths selected by different variable selection algorithms from spectra and texture data of Vis-SWNIR and LWNIR regions, respectively. (**a**) Spectrum in Vis-SWNIR region, (**b**) energy parameters in Vis-SWNIR region, (**c**) spectrum in LWNIR region, and (**d**) contrast parameters in LWNIR region. Note: red points, black points, and green points represent the variables retained by VCPA, IRIV, and mVCPA-IRIV in the spectrum or texture parameters, respectively. Orange points represent the bands jointly selected by all three algorithms.

**Figure 8 foods-11-01727-f008:**
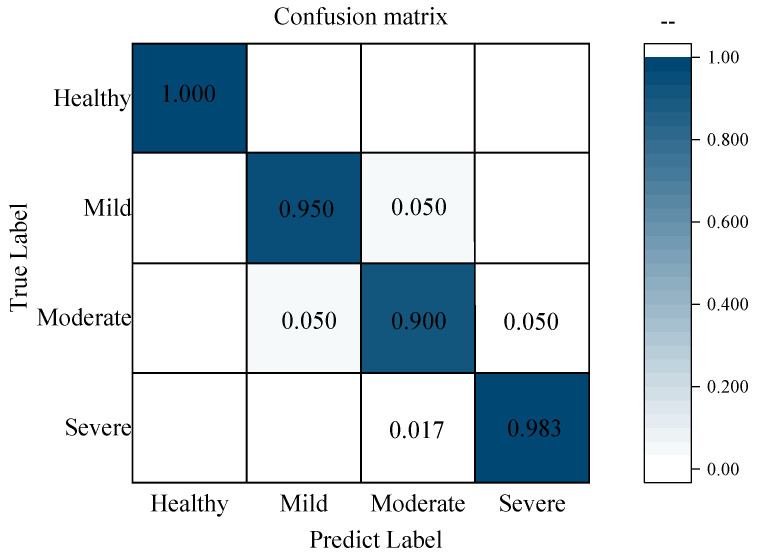
Confusion matrix of overall prediction results for all samples.

**Table 1 foods-11-01727-t001:** Catalase (CAT) activity value of maize with different moldy levels.

Moldy Level	Mean mL/(h × g)	Standard Deviation mL/(h × g)
Healthy level	0	0
Mild level	1.57	0.13
Moderate level	1.91	0.09
Severe level	2.24	0.12

**Table 2 foods-11-01727-t002:** Division of maize with different moldy levels in the calibration and prediction sets.

Moldy Level		Calibration Set			Prediction Set	
	Num	Mean	Standard Deviation	Num	Mean	Standard Deviation
Healthy level	35	0	0	15	0	0
Mild level	35	1.58	0.17	15	1.58	0.14
Moderate level	35	1.94	0.18	15	1.96	0.20
Severe level	35	2.20	0.14	15	2.13	0.24

**Table 3 foods-11-01727-t003:** The classification performance of the SVM models established by different preprocessing methods.

Classifier	Sensor	Spectral Preprocessing Method	Calibration Set Accuracy (%)	Prediction Set Accuracy (%)
SVM	Vis-SWNIR	smooth-msc	84.44	88.33
		smooth-detrend	86.67	88.33
		smooth-center	85.56	90.00
	LWNIR	smooth-msc	90.56	86.67
		smooth-detrend	90.56	88.33
		smooth-center	91.11	85.00

**Table 4 foods-11-01727-t004:** The classification results of the pixel-level fusion of spectral and texture information.

Sensor	Data Source		Calibration Set Accuracy (%)	Prediction Set Accuracy (%)
	Spectra	Texture		
Vis-SWNIR	smooth-detrend	contrast	85.56	86.67
		correction	85.56	86.67
		energy	92.22	90.00
		homogeneity	85.56	86.67
LWNIR	smooth-detrend	contrast	92.22	90.00
		correction	92.22	88.33
		energy	97.78	85.00
		homogeneity	92.22	88.33

**Table 5 foods-11-01727-t005:** The classification results of the feature-level fusion of spectral and texture information.

Integration Method	Sensor	Data Source	Variable Selection Algorithm	Characteristic Number		Calibration Set Accuracy (%)	Prediction Set Accuracy (%)
				Spectra	Texture		
Feature-level fusion	Vis-SWNIR	smooth-detrend energy	VCPA	9	12	94.44	93.33
			IRIV	21	28	97.78	95.00
			mVCPA-IRIV	28	39	93.89	91.67
	LWNIR	smooth-detrend contrast	VCPA	12	12	96.67	90.00
			IRIV	13	35	100.00	83.33
			mVCPA-IRIV	17	41	99.44	91.97

## Data Availability

The data that support the findings of this study are available upon request from the authors.

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
