# Peer review of "Identification of Maize with Different Moldy Levels Based on Catalase Activity and Data Fusion of Hyperspectral Images"

_foods, 2022, doi:10.3390/foods11121727_

Round 1

Reviewer 1 Report

foods-1737050-peer-review-v1

The authors presented a study to evaluate the use of hyperspectral imaging to detect maize moldy texture. The manuscript presents a degree of novelty and should following comments are expressed.

INTRODUCTION

Line 32-34: Provide values of moisture content to compare between different crops.

Introduction in general is too long. It needs to be reduced and be more concise and focused.

Materials and Methods

Line 150 “All maize kernels  were surface sterilized in 2% sodium hypochlorite solution soaked for 5 minutes…”. Replace with “All maize kernels were surface sterilized by soaking in 2% sodium hypochlorite solution for 5 minutes…”.

The authors need to state the source of the Aspergillus flavus (BNCC142801).

The authors need to add examples of RGB images for healthy, and moldy samples.

Line 180: “, respectively” instead of “respectively”.

2.3. Add a reference for the method of determining the CAT activity.

Lines 300-303: The authors listed 3 feature selection methods whereas 4 methods were claimed at the beginning.      

Results and Discussion

The authors well explained the results in all sections. However, more comparison with previous studies is needed in sections 3.3, 3.4, 3.5, , and 3.6

Conclusions 

The authors need to add some general statistics that can briefly summarize the study achievements.  

Reviewer 2 Report

The manuscript is written with clear understanding of the project addressed. However, there are major concerns that need to be addressed to enhance the quality of the manuscript. My specific comments are as follows:

Abstract:

Elaborate more on the methods used in this study.

Introduction:

Page2Line71: “Machine vision detection mainly adopts machine learning algorithms to extract features from kernel images…” Add citation

P3L100: “Previous studies have shown that both the spectral ranges of Vis-SWNIR and LWNIR can be used to distinguish the moldy level of maize…” Elaborate more on the previous studies

P3L120: “Catalase (CAT) is a metabolite in the growth of Aspergillus flavus and other microorganisms.” Add citation

P3133: “…object..” change to ‘objective’

Based on your objectives, please compare how your study is different from those that have already been published

Materials and Methods:

How many fruit samples for overall study?

Combine Section 2.4 into 2.5

P7L256: “Hence, moving smooth, multiple scattering correction (msc), detrend, and mean centralization (center) were used in this study.” Explain in details the steps/methodology for data preprocessing

P7L275: “Pixel-level fusion was simply merging the data information of different sources…” Add citation

P7L284: “Feature-level fusion was to extract features from a single data block using…” Add citation

Combine Section 2.8 into 2.7

Section 2.9 should be placed before L339 of Section 2.8

Results and Discussion:

P12L399: “It should be pointed out that the samples of calibration and prediction sets were no longer changed and no sample was used in both calibration and prediction sets.” What does it means by this statement? No sample was used?

Table 2 is the result for SVM only?

Table 3: How about the results from other classifers, RF and KNN?

P15L493: “According to previous studies, wavelengths around 768 and 857 nm were classified as effective bands for mycorrhizal infestation of maize seeds…” Which studies?

Conclusions:

Elaborate more on the main findings from your study.

Add on recommendation for future studies.

General comments:

Please check the reference styles and grammar of the manuscript.

Reviewer 3 Report

This manuscript is extremely scientifically interesting, and the topic is very important from the point of view of food production and its safety. Thematically, this manuscript fits the Foods journal. The authors must take into account the suggestions indicated:

·         The presented research results were obtained under model conditions, without taking into account the microflora naturally contaminating maize grains in various environmental and climatic conditions. All maize kernels were surface chemically sterilized. In my opinion, these aspects should be absolutely discussed when discussing the results, with reference to the pilot nature of these studies.

·         The research does not take into account the applied agrotechnical practices practiced by the harvesting and storage of maize. This greatly restricts the applicability of the results obtained. In my opinion, this should also be taken into account when discussing the results.

·         The research was conducted on a small number of experiments, trials and their replications. It is extremely important from a statistical point of view, determining the repeatability of the experimental results, the reproducibility and accuracy of the measurement results. This should also be considered when discussing the results.

Round 2

Reviewer 1 Report

The authors have addressed the comments and the manuscript can be published in its current form.